# Classify and Generate Reciprocally: Simultaneous Positive-Unlabelled Learning and Conditional Generation with Extra Data

## Abstract

The scarcity of class-labeled data is a ubiquitous bottleneck in a wide range of machine learning problems. While abundant unlabeled data normally exist and provide a potential solution, it is extremely challenging to exploit them. In this paper, we address this problem by leveraging Positive-Unlabeled (PU) classification and the conditional generation with extra unlabeled data *simultaneously*, both of which aim to make full use of agnostic unlabeled data to improve classification and generation performance. In particular, we present a novel training framework to jointly target both PU classification and conditional generation when exposing to extra data, especially out-of-distribution unlabeled data, by exploring the interplay between them: 1) enhancing the performance of PU classifiers with the assistance of a novel Conditional Generative Adversarial Network (CGAN) that is robust to noisy labels, 2) leveraging extra data with predicted labels from a PU classifier to help the generation. Our key contribution is a Classifier-Noise-Invariant Conditional GAN (CNI-CGAN) that can learn the clean data distribution from noisy labels predicted by a PU classifier. Theoretically, we proved the optimal condition of CNI-CGAN and experimentally, we conducted extensive evaluations on diverse datasets, verifying the simultaneous improvements on both classification and generation.

## 1 Introduction

Existing machine learning methods, particularly deep learning models, typically require big data to pursue remarkable performance. For instance, conditional deep generative models are able to generate high-fidelity and diverse images, but they have to rely on vast amounts of labeled data (Lucic et al., 2019). Nevertheless, it is often laborious or impractical to collect large-scale accurate class-labeled data in real-world scenarios, and thus the label scarcity is ubiquitous. Under such circumstances, the performance of classification and conditional generation (Mirza & Osindero, 2014) drops significantly (Lucic et al., 2019). At the same time, diverse unlabeled data are available in enormous quantities, and therefore a key issue is how to take advantage of the extra data to enhance the conditional generation or classification.

Within the unlabeled data, both in-distribution and out-of-distribution data exist, where in-distribution data conform to the distribution of the labeled data while out-of-distribution data do not. Our key insight is to harness the out-of-distribution data. In the generation with extra data, most related works focused on the in-distribution data (Lucic et al., 2019; Gui et al., 2020; Donahue & Simonyan, 2019). When it comes to the out-of-distribution data, the majority of existing methods (Noguchi & Harada, 2019; Yamaguchi et al., 2019; Zhao et al., 2020) attempted to forcibly train generative models on a large amount of unlabeled data, and then transferred the learned knowledge of the pre-trained generator to the in-distribution data. In classification, a common setting to utilize unlabeled data is semi-supervised learning (Miyato et al., 2018; Sun et al., 2019; Berthelot et al., 2019), which usually assumes that the unlabeled and labeled data come from the same distribution, ignoring their distributional mismatch. In contrast, Positive and Unlabeled (PU) Learning (Bekker & Davis, 2020; Kiryo et al., 2017) is an elegant way of handling this under-studied problem, where a model has the only access to positive samples and unlabeled data. Therefore, it is possible to utilize pseudo labels predicted by a PU classifier on unlabeled data to guide the conditional gen-

eration. However, the predicted signals from the classifier tend to be noisy. Although there are a flurry of papers about learning from noisy labels for classification (Tsung Wei Tsai, 2019; Ge et al., 2020; Guo et al., 2019), to our best knowledge, no work has considered to leverage the noisy labels seamlessly in the joint classification and generation. Additionally, another work (Hou et al., 2018) leveraged GANs to recover both positive and negative data distribution to step away from overfitting, but they never considered the noise-invariant generation or their mutual improvement. The generative-discriminative complementary learning (Xu et al., 2019) was investigated in weakly supervised learning, but we are the first attempt to tackle the (Multi-) Positive and Unlabeled learning setting while developing the method of noise-invariant generation from noisy labels. Please refer to Section 5 for the discussion about more related works.

In this paper, we focus on the mutual benefits of conditional generation and PU classification, when we are only accessible to little class-labeled data, but extra unlabeled data, including out-of-distribution data, can be available. Firstly, a parallel non-negative multi-class PU estimator is derived to classify both the positive data of all classes and the negative data. Then we design a Classifier-Noise-Invariant Conditional Generative Adversarial Network (CNI-CGAN) that is able to learn the clean data distribution on all unlabeled data with noisy labels provided by the PU classifier. Simultaneously, we also leverage our CNI-CGAN to enhance the performance of the PU classification through data augmentation, demonstrating a reciprocal benefit for both generation and classification. We provide the theoretical analysis on the optimal condition of our CNI-CGAN and conduct extensive experiments to verify the superiority of our approach.

## 2 OUR METHOD

### 2.1 POSITIVE-UNLABELED LEARNING

**Traditional Binary Positive-Unlabeled Problem Setting** Let $X \in \mathbb{R}^d$ and $Y \in \{\pm 1\}$ be the input and output variables and $p(x, y)$ is the joint distribution with marginal distribution $p_p(x) = p(x|Y = +1)$ and $p_n(x) = p(x|Y = -1)$. In particular, we denote $p(x)$ as the distribution of unlabeled data. $n_p$, $n_n$ and $n_u$ are the amount of positive, negative and unlabeled data, respectively.

**Parallel Non-Negative PU Estimator** Vanilla PU learning (Bekker & Davis, 2020; Kiryo et al., 2017; Du Plessis et al., 2014; 2015) employs unbiased and consistent estimator. Denote $g_\theta : \mathbb{R}^d \to \mathbb{R}$ as the score function parameterized by $\theta$, and $\ell : \mathbb{R} \times \{\pm 1\} \to \mathbb{R}$ as the loss function. The risk of $g_\theta$ can be approximated by its empirical version denoted as $\widehat{R}_{\mathrm{pn}}(g_\theta)$:

$$\widehat{R}_{\mathrm{pn}}(g_\theta) = \pi_{\mathrm{p}} \widehat{R}_{\mathrm{p}}^+(g_\theta) + \pi_{\mathrm{n}} \widehat{R}_{\mathrm{n}}^-(g_\theta), \tag{1}$$

where $\pi_p$ represents the class prior probability, i.e. $\pi_p = P(Y = +1)$ with $\pi_p + \pi_n = 1$. In addition, $\widehat{R}_{\mathrm{p}}^+(g_\theta) = \frac{1}{n_p} \sum_{i=1}^{n_{\mathrm{p}}} \ell\left(g_\theta\left(x_i^{\mathrm{p}}\right), +1\right)$ and $\widehat{R}_{\mathrm{n}}^-(g_\theta) = \frac{1}{n_n} \sum_{i=1}^{n_{\mathrm{n}}} \ell\left(g_\theta\left(x_i^{\mathrm{n}}\right), -1\right)$.

As negative data $x^n$ are unavailable, a common strategy is to offset $R_{\mathrm{n}}^-(g_\theta)$. We also know that $\pi_{\mathrm{n}} p_{\mathrm{n}}(x) = p(x) - \pi_{\mathrm{p}} p_{\mathrm{p}}(x)$, and hence $\pi_{\mathrm{n}} \widehat{R}_{\mathrm{n}}^-(g_\theta) = \widehat{R}_{\mathrm{u}}^-(g_\theta) - \pi_{\mathrm{p}} \widehat{R}_{\mathrm{p}}^-(g_\theta)$. Then the resulting unbiased risk estimator $\widehat{R}_{\mathrm{pu}}(g_\theta)$ can be formulated as:

$$\widehat{R}_{\mathrm{pu}}(g_\theta) = \pi_{\mathrm{p}} \widehat{R}_{\mathrm{p}}^+(g_\theta) - \pi_{\mathrm{p}} \widehat{R}_{\mathrm{p}}^-(g_\theta) + \widehat{R}_{\mathrm{u}}^-(g_\theta), \tag{2}$$

where $\widehat{R}_{\mathrm{p}}^-(g_\theta) = \frac{1}{n_p} \sum_{i=1}^{n_{\mathrm{p}}} \ell\left(g_\theta\left(x_i^{\mathrm{p}}\right), -1\right)$ and $\widehat{R}_{\mathrm{u}}^-(g_\theta) = \frac{1}{n_u} \sum_{i=1}^{n_{\mathrm{u}}} \ell\left(g_\theta\left(x_i^{\mathrm{u}}\right), -1\right)$. The advantage of this unbiased risk minimizer is that the optimal solution can be easily obtained if $g$ is linear in $\theta$. However, in real scenarios we tend to leverage more flexible models $g_\theta$, e.g., deep neural networks. This strategy will push the estimator to a point where it starts to suffer from overfitting. Hence, we decide to utilize *non-negative risk* (Kiryo et al., 2017) for our PU learning, which has been verified in (Kiryo et al., 2017) to allow deep neural network to mitigate overfitting. The non-negative PU estimator is formulated as:

$$\widehat{R}_{\mathrm{pu}}(g_\theta) = \pi_{\mathrm{p}} \widehat{R}_{\mathrm{p}}^+(g_\theta) + \max\left\{0, \widehat{R}_{\mathrm{u}}^-(g_\theta) - \pi_{\mathrm{p}} \widehat{R}_{\mathrm{p}}^-(g_\theta)\right\}. \tag{3}$$

In pursue of the parallel implementation of $\widehat{R}_{\mathrm{pu}}(g_\theta)$, we replace $\max\left\{0, \widehat{R}_{\mathrm{u}}^-(g_\theta) - \pi_{\mathrm{p}} \widehat{R}_{\mathrm{p}}^-(g_\theta)\right\}$ with its lower bound $\frac{1}{N} \sum_{i=1}^{N} \max\left\{0, \widehat{R}_{\mathrm{u}}^-(g_\theta; \mathcal{X}_u^i) - \pi_{\mathrm{p}} \widehat{R}_{\mathrm{p}}^-(g_\theta; \mathcal{X}_p^i)\right\}$ where $\mathcal{X}_u^i$ and $\mathcal{X}_p^i$ denote as the unlabeled and positive data in the $i$-th mini-batch, and $N$ is the number of batches.

**From Binary PU to Multi-PU Learning** Previous PU learning focuses on learning a classifier from positive and unlabeled data, and cannot easily be adapted to $K + 1$ multi-classification tasks where $K$ represents the number of classes in the positive data. Multi-Positive and Unlabeled learning (Xu et al., 2017) was ever developed, but the proposed algorithm may not allow deep neural networks. Instead, we extend binary PU learning to multi-class version in a straightforward way by additionally incorporating cross entropy loss on all the positive data with labels for different classes. More precisely, we consider the $K + 1$-class classifier $f_\theta$ as a score function $f_\theta = \left( f_\theta^1(x), \dots, f_\theta^{K+1}(x) \right)$. After the *softmax* function, we select the first $K$ positive data to construct cross-entropy loss $\ell^{\mathrm{CE}}$, i.e., $\ell^{\mathrm{CE}}(f_\theta(x), y) = \log \sum_{j=1}^{K+1} \exp \left( f_\theta^j(x) \right) - f_\theta^y(x)$ where $y \in [K]$. For the PU loss, we consider the composite function $h(f_\theta(x)) : \mathbb{R}^d \to \mathbb{R}$ where $h(\cdot)$ conducts a logit transformation on the accumulative probability for the first $K$ classes, i.e., $h(f_\theta(x)) = \ln(\frac{p}{1-p})$ in which $p = \sum_{j=1}^K \exp \left( f_\theta^j(x) \right) / \sum_{j=1}^{K+1} \exp \left( f_\theta^j(x) \right)$. The final mini-batch risk of our PU learning can be presented as:

$$\widetilde{R}_{\mathrm{pu}}(f_\theta; \mathcal{X}^i) = \pi_{\mathrm{p}} \widehat{R}_{\mathrm{p}}^+(h(f_\theta); \mathcal{X}_p^i) + \max \left\{ 0, \widehat{R}_{\mathrm{u}}^-(h(f_\theta); \mathcal{X}_u^i) - \pi_{\mathrm{p}} \widehat{R}_{\mathrm{p}}^-(h(f_\theta); \mathcal{X}_p^i) \right\} + \widehat{R}_{\mathrm{p}}^{\mathrm{CE}}(f_\theta; \mathcal{X}_p^i), \tag{4}$$

where $\widehat{R}_{\mathrm{p}}^{\mathrm{CE}}(f_\theta; \mathcal{X}_p^i) = \frac{1}{n_p} \sum_{i=1}^{n_{\mathrm{p}}} \ell^{\mathrm{CE}} \left( f_\theta \left( x_i^{\mathrm{p}} \right), y \right)$.

## 2.2 CLASSIFIER-NOISE-INVARIANT CONDITIONAL GENERATIVE ADVERSARIAL NETWORK (CNI-CGAN)

To leverage extra data, i.e., all unlabeled data, to benefit the generation, we deploy our conditional generative model on all data with pseudo labels predicted by our PU classifier. However, these predicted labels tend to be noisy, reducing the reliability of the supervision signals and thus worsening the performance of the conditional generative model. Besides, the noise depends on the accuracy of the given PU classifier. To address this issue, we focus on developing a novel noise-invariant conditional GAN that is robust to noisy labels provided by a specified classifier, e.g. a PU classifier. We call our method Classifier-Noise-Invariant Conditional Generative Adversarial Network (CNI-CGAN) and the architecture is depicted in Figure 1. In the following, we elaborate on each part of it.

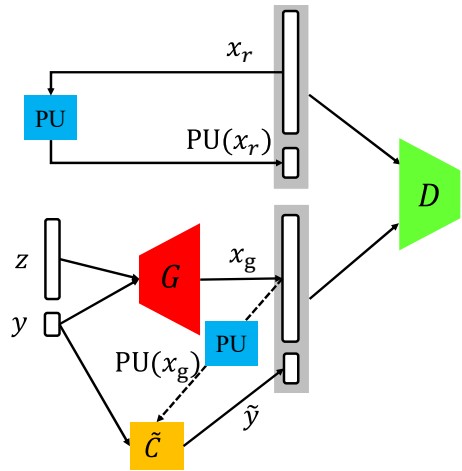

Figure 1: Model architecture of our Classifier-Noise-Invariant Conditional GAN (CNI-CGAN). The output $x_g$ of the conditional generator $G$ is paired with a noisy label $\tilde{y}$ corrupted by the PU-dependent confusion matrix $\tilde{C}$. The discriminator $D$ distinguishes between whether a given labeled sample comes from the real data $(x_r, PU_\theta(x_r))$ or generated data $(x_g, \tilde{y})$.

**Principle of the Design of CNI-CGAN**

Albeit being noisy, the pseudo labels given by the PU classifier still provide rich information that we can exploit. The key is to take the noise generation mechanism into consideration during the generation. We denote the real data as $x_r$ and the predicted hard label through the PU classifier as $PU_\theta(x_r)$, i.e., $PU_\theta(x_r) = \arg\max_i f_\theta^i(x_r)$, as displayed in Figure 1. We let the generator "imitate" the noise generation mechanism to generate pseudo labels for the labeled data. With both pseudo and real labels, we can leverage the PU classifier $f_\theta$ to estimate a confusion matrix $\tilde{C}$ to model the label noise from the classifier. During the generation, a real label $y$, while being fed into the generator $G$, will also be polluted by $\tilde{C}$ to compute a noisy label $\tilde{y}$, which then will be combined with the generated fake sample $x_g$ for the following discrimination. Finally, the discriminator $D$ will distinguish the real samples $[x_r, PU_\theta(x_r)]$ out of fake samples $[x_g, \tilde{y}]$. Overall, the noise "generation" mechanism from both sides can be balanced.

**Estimation of** $\tilde{C}$ The key in the design of $\tilde{C}$ is to estimate the label noise of the pre-trained PU classifier by considering all the samples of each class. More specifically, the confusion matrix $\tilde{C}$ is $k + 1$ by $k + 1$ and each entry $\tilde{C}_{ij}$ represents the probability of a generated sample $x_g$, given a label $i$, being classified as class $j$ by the PU classifier. Mathematically, we denote $\tilde{C}_{ij}$ as:

$$\tilde{C}_{ij} = P(PU_\theta(x_g) = j | y = i) = \mathbb{E}_z[\mathbb{I}_{\{PU_\theta(x_g) = j | y = i\}}], \tag{5}$$

where $x_g = G(z, y = i)$ and $\mathbb{I}$ is the indicator function. Owing to the stochastic optimization nature when training deep neural networks, we incorporate the estimation of $\tilde{C}$ in the processing of training by *Exponential Moving Average (EMA) method*. This choice can balance the utilization of information from previous training samples and the updated PU classifier to estimate $\tilde{C}$. We formulate the update of $\tilde{C}^{(l+1)}$ in the $l$-th mini-batch as follows:

$$\tilde{C}^{(l+1)} = \lambda \tilde{C}^{(l)} + (1 - \lambda) \Delta^{\tilde{C}}_{\mathcal{X}_l}, \tag{6}$$

where $\Delta^{\tilde{C}}_{\mathcal{X}_l}$ denotes the incremental change of $\tilde{C}$ on the current $l$-th mini-batch data $\mathcal{X}_l$ via Eq. 5. $\lambda$ is the averaging coefficient in EMA.

**Theoretical Guarantee of Clean Data Distribution** Firstly, we denote $\mathcal{O}(x)$ as the oracle class of sample $x$ from an oracle classifier $\mathcal{O}(\cdot)$. Let $\pi_i, i = 1, ..., K + 1$, be the class-prior probability of the class $i$ in the multi-positive unlabeled setting. Theorem 1 proves the optimal condition of CNI-CGAN to guarantee the convergence to the clean data distribution. The proof is provided in Appendix A.

**Theorem 1.** *(Optimal Condition of CNI-CGAN) Let $P^g$ be a probabilistic transition matrix where $P^g_{ij} = P(\mathcal{O}(x_g) = j | y = i)$ indicates the probability of sample $x_g$ with the oracle label $j$ generated by $G$ with the initial label $i$. We assume that the conditional sample space of each class is disjoint with each other, then*

*(1) $P^g$ is a permutation matrix if the generator $G$ in CNI-CGAN is optimal, with the permutation, compared with an identity matrix, only happens on rows $\mathbf{r}$ where corresponding $\pi_r, r \in \mathbf{r}$ are equal.*

*(2) If $P^g$ is an identity matrix and the generator $G$ in CNI-CGAN is optimal, then $p^r(x, y) = p^g(x, y)$ where $p^r(x, y)$ and $p^g(x, y)$ are the real and the generating joint distribution, respectively.*

Briefly speaking, CNI-CGAN can learn the clean data distribution if $P^g$ is an identity matrix. More importantly, the method we elaborate till now has already guaranteed $P_g$ as a permutation matrix, which is very close to an identity one. We need an additional constraint, although the permutation happens only when same class-prior probabilities exist.

**The Auxiliary Loss** The optimal $G$ in CNI-CGAN can only guarantee that $p^g(x, y)$ is close to $p^r(x, y)$ as the optimal permutation matrix $P^g$ is close to the identity matrix. Hence in practice, to ensure that we can exactly learn an identity matrix for $P^g$ and thus achieve the clean data distribution, we introduce an auxiliary loss to encourage a larger trace of $P^g$, i.e., $\sum_{i=1}^{K+1} P(\mathcal{O}(x_g) = i) | y = i)$. As $\mathcal{O}(\cdot)$ is intractable, we approximate it by the current PU classifier $PU_\theta(x_g)$. Then we obtain the auxiliary loss $\ell_{\text{aux}}$:

$$\ell_{\text{aux}}(z, y) = \max\{\kappa - \frac{1}{K + 1} \sum_{i=1}^{K+1} \mathbb{E}_z(\mathbb{I}_{\{PU_\theta(x_g) = i | y = i\}}), 0\}, \tag{7}$$

where $\kappa \in (0, 1)$ is a hyper-parameter. With the support of auxiliary loss, $P^g$ has the tendency to converge to the identity matrix where CNI-CGAN can learn the clean data distribution even in the presence of noisy labels.

**Comparison with RCGAN (Thekumparampil et al., 2018; Kaneko et al., 2019)** The theoretical property of CNI-CGAN has a major advantage over existing Robust CGAN (RC-GAN) (Thekumparampil et al., 2018; Kaneko et al., 2019), for which the optimal condition can only be achieved when the label confusion matrix is known *a priori*. Although heuristics can be employed, such as RCGAN-U (Thekumparampil et al., 2018), to handle the unknown label noise setting, these approaches still lack the theoretical guarantee to converge to the clean data distribution.

To guarantee the efficacy of our approach, one implicit and mild assumption is that our PU classifier will not overfit on the training data, while our non-negative estimator helps to ensure that it as

explained in Section 2.1. To further clarify the optimization process of CNI-CGAN, we elaborate the training steps of $D$ and $G$, respectively.

**D-Step:** We train $D$ on an adversarial loss from both the real data and the generated $(x_g, \tilde{y})$, where $\tilde{y}$ is corrupted by $\tilde{C}$. $\tilde{C}_y$ denotes the $y$-th row of $\tilde{C}$. We formulate the loss of $D$ as:

$$\max_{D \in \mathcal{F}} \mathbb{E}_{x \sim p(x)} [\phi(D(x, PU_\theta(x)))] + \mathbb{E}_{\substack{z \sim P_Z, y \sim P_Y \\ \tilde{y}|y \sim \tilde{C}_y}} [\phi(1 - D(G(z, y), \tilde{y}))], \tag{8}$$

where $\mathcal{F}$ is a family of discriminators and $P_Z$ is the distribution of latent space vector $z$, e.g., a Normal distribution. $P_Y$ is a discrete uniform distribution on $[K + 1]$ and $\phi$ is the measuring function.

**G-Step:** We train $G$ additionally on the auxiliary loss $\ell_{\mathrm{aux}}(z, y)$ as follows:

$$\min_{G \in \mathcal{G}} \mathbb{E}_{\substack{z \sim P_Z, y \sim P_Y \\ \tilde{y}|y \sim \tilde{C}_y}} [\phi(1 - D(G(z, y), \tilde{y})) + \beta \ell_{\mathrm{aux}}(z, y)], \tag{9}$$

where $\beta$ controls the strength of auxiliary loss and $\mathcal{G}$ is a family of generators. In summary, our CNI-CGAN conducts $K + 1$ classes generation, which can be further leveraged to benefit the $K + 1$ PU classification via data augmentation.

---

**Algorithm 1** Alternating Minimization for PU Learning and Classifier-Noise-Invariant Generation.

---

**Input**: Training data $(\mathcal{X}_p, \mathcal{X}_u)$. Batch size $M$ and hyper-parameter $\beta > 0$, $\lambda, \kappa \in (0, 1)$. $L_0$ and $L \in N^+$. Initializing $\tilde{C}^{(1)}$ as identity matrix. Number of batches $N$ during the training.
**Output**: Model parameter for generator $G$, and $\theta$ for the PU classifier $f_\theta$.

1: / * *Pre-train PU classifier* $f_\theta$ * /
2: **for** $i = 1$ to $N$ **do**
3:     Update $f_\theta$ by descending its stochastic gradient of $\widetilde{R}_{\mathrm{pu}}(f_\theta; \mathcal{X}^i)$ via Eq. 4.
4: **end for**
5: **repeat**
6:     / * *Update CNI-CGAN* * /
7:     **for** $l = 1$ to $L$ **do**
8:         Sample $\{\mathbf{z}_1, ..., \mathbf{z}_M\}$, $\{\mathbf{y}_1, ..., \mathbf{y}_M\}$ and $\{\mathbf{x}_1, ..., \mathbf{x}_M\}$ from $P_Z$, $P_Y$ and all training data, respectively, and then sample $\{\tilde{\mathbf{y}}_1, ..., \tilde{\mathbf{y}}_M\}$ through the current $\tilde{C}^{(l)}$. Then, update the discriminator $D$ by ascending its stochastic gradient of

$$\frac{1}{M} \sum_{i=1}^{M} [\phi(D(\mathbf{x}_i, PU_\theta(\mathbf{x}_i)))] + \phi(1 - D(G(\mathbf{z}_i, \mathbf{y}_i), \tilde{\mathbf{y}}_i))].$$

9:         Sample $\{\mathbf{z}_1, ..., \mathbf{z}_M\}$ and $\{\mathbf{y}_1, ..., \mathbf{y}_M\}$ from $P_Z$ and $P_Y$, and then sample $\{\tilde{\mathbf{y}}_1, ..., \tilde{\mathbf{y}}_M\}$ through the current $\tilde{C}^{(l)}$. Update the generator $G$ by descending its stochastic gradient of

$$\frac{1}{M} \sum_{i=1}^{M} [\phi(1 - D(G(\mathbf{z}_i, \mathbf{y}_i), \tilde{\mathbf{y}}_i)) + \beta \ell_{\mathrm{aux}}(\mathbf{y}_i, \mathbf{z}_i)].$$

10:         **if** $l \geq L_0$ **then**
11:             Compute $\Delta_{\mathcal{X}_l}^{\tilde{C}} = \frac{1}{M} \sum_{i=1}^{M} \mathbb{I}_{\{PU_\theta(G(\mathbf{z}_i, \mathbf{y}_i))|\mathbf{y}_i\}}$ via Eq. 5, and then estimate $\tilde{C}$ by
$$\tilde{C}^{(l+1)} = \lambda \tilde{C}^{(l)} + (1 - \lambda) \Delta_{\mathcal{X}_l}^{\tilde{C}}.$$

12:         **end if**
13:     **end for**
14:     / * *Update PU classifier via Data Augmentation* * /
15:     Sample $\{\mathbf{z}_1, ..., \mathbf{z}_M\}$ and $\{\mathbf{y}_1, ..., \mathbf{y}_M\}$ from $P_Z$ and $P_Y$, respectively, and then update the PU classifier $f_\theta$ by descending its stochastic gradient of

$$\frac{1}{M} \sum_{i=1}^{M} \ell^{\mathrm{CE}}(f_\theta(G(\mathbf{z}_i, \mathbf{y}_i)), \mathbf{y}_i).$$

16: **until** convergence

---

## 3 ALGORITHM

Firstly, we obtain a PU classifier $f_\theta$ trained on multi-positive and unlabeled dataset with the parallel non-negative estimator derived in Section 2.1. Then we train our CNI-CGAN, described in Section 2.2, on all data with pseudo labels predicted by the pre-trained PU classifier. As our CNI-CGAN is robust to noisy labels, we leverage the data generated by CNI-CGAN to conduct data augmentation to improve the PU classifier. Finally, we implement the joint optimization for the training of CNI-CGAN and the data augmentation of the PU classifier. We summarize the procedure in Algorithm 1 and provide more details in Appendix C.

**Computational Cost Analysis** In the implementation of our CNI-CGAN, we need to additionally estimate $\tilde{C}$, a $(K+1) \times (K+1)$ matrix. The computational cost of this small matrix is negligible compared with the updating of discriminator and generator networks, although the estimation of $\tilde{C}$ is crucial.

**Simultaneous Improvement on PU Learning and Generation with Extra Data** From the perspective of PU classification, due to the theoretical guarantee from Theorem 1, CNI-CGAN is capable of learning a clean data distribution out of noisy pseudo labels predicted by the pre-trained PU classifier. Hence, the following data augmentation has the potential to improve the generalization of PU classification regardless of the specific form of the PU estimator. From the perspective of generation with extra data, the predicted labels on unlabeled data from the PU classifier can provide CNI-CGAN with more supervised signals, thus further improving the quality of generation. Due to the joint optimization, both the PU classification and conditional generative models are able to improve each other reciprocally, as demonstrated in the following experiments.

## 4 EXPERIMENT

**Experimental Setup** We perform our approaches and several baselines on MNIST, Fashion-MNIST and CIFAR-10. We select the first 5 classes on MNIST and 5 non-clothes classes on Fashion-MNIST, respectively, for $K+1$ classification ($K=5$). To verify the consistent effectiveness of our method in the standard binary PU setting, we pick the 4 categories of transportation tools in CIFAR-10 as the one-class positive dataset. As for the baselines, the first is *CGAN-P*, where a Vanilla CGAN (Mirza & Osindero, 2014) is trained only on limited positive data. Another natural baseline is *CGAN-A* where a Vanilla CGAN is trained on all data with labels given by the PU classifier.

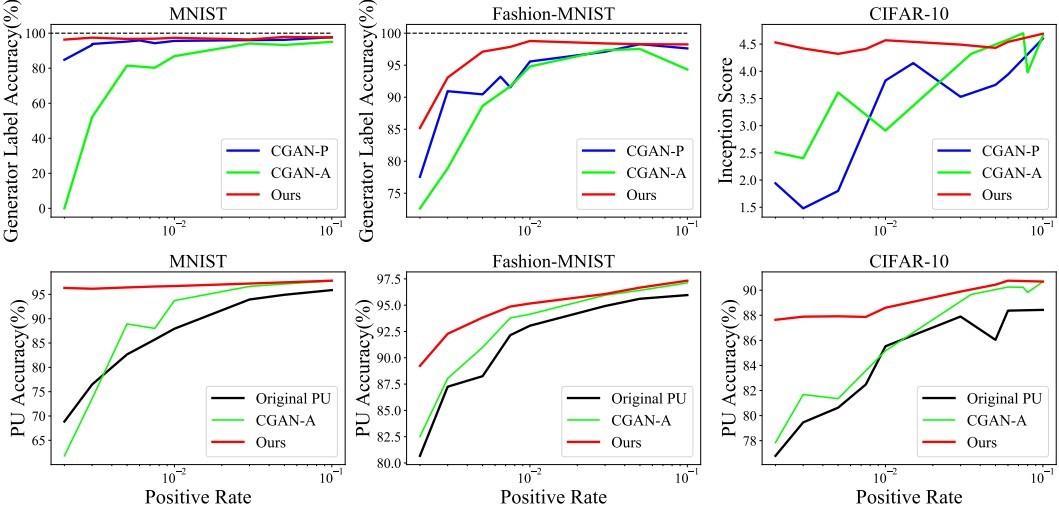

Figure 2: Generation and classification performance of CGAN-P, CGAN-A and Ours on three datasets. Results of CGAN-P (blue lines) on PU accuracy do not exist since CGAN-P generates only $K$ classes data rather than $K+1$ categories that the PU classifier needs.

The last baseline is RCGAN-U (Thekumparampil et al., 2018) where the confusion matrix is totally learnable while training. For fair comparisons, we choose the same GAN architecture. Through a line search of hyper-parameters, we choose $\kappa$ as 0.75, $\beta$ as 5.0 and $\lambda = 0.99$ across all the datasets. We set $L_0$ as 5 in Algorithm 1. More details about hyper-parameters can be found in Appendix D.

**Evaluation Metrics** For MNIST and Fashion-MNIST, we mainly use *Generator Label Accuracy* (Thekumparampil et al., 2018) and *PU Accuracy* to evaluate the quality of generated images. *Generator Label Accuracy* compares specified $y$ from CGANs to the true class of the generated examples through a pre-trained (almost) oracle classifier $f$. In experiments, we pre-trained two $K+1$ classifiers with 99.28% and 98.23% accuracy on the two datasets, respectively. Additionally, the increased *PU Accuracy* measures the closeness between generated data distribution and test (almost real) data distribution for the PU classification, serving as a key indicator to reflect the quality of generated images. For CIFAR 10, we use both *Inception Score* (Salimans et al., 2016) to evaluate the quality of the generated samples, and the increased *PU Accuracy* to quantify the improvement of generated samples on the PU classification.

## 4.1 GENERATION AND CLASSIFICATION PERFORMANCE

We set the whole training dataset as the unlabeled data and select certain amount of positive data with the ratio of *Positive Rate*. Figure 2 presents the trend of Generator Label Accuracy, Inception Score and PU Accuracy as the Positive Rate increases. It turns out that CNI-CGAN outperforms CGAN-P and CGAN-A consistently especially when the Positive Rate is small, i.e. little positive data. Remarkably, our approach enhances the PU accuracy greatly when exposed to low positive rates, while CGAN-A even worsens the original PU classifier sometimes in this scenario due to the existence of too much label noise given by a less accurate PU classifier. Meanwhile, when more supervised positive data are given, the PU classifier generalizes better and then provides more accurate labels, conversely leading to more consistent and better performance for all methods. Besides, note that even though the CGAN-P achieves comparable generator label accuracy on MNIST, it results in a lower Inception Score. We demonstrate this in Appendix D.

Table 1: PU classification accuracy of RCGAN-U and Ours across three datasets. Final PU accuracy represents the accuracy of PU classifier after the data augmentation.

| Final PU Accuracy \ Positive Rates (%) | | 0.2% | 0.5% | 1.0% | 10.0% |
|---|---|---|---|---|---|
| | Original PU | 68.86 | 76.75 | 86.94 | 95.88 |
| MNIST | RCGAN-U | 87.95 | 95.24 | 95.86 | **97.80** |
| | Ours | **96.33** | **96.43** | **96.71** | **97.82** |
| | Original PU | 80.68 | 88.25 | 93.05 | 95.99 |
| Fashion-MNIST | RCGAN-U | 89.21 | 92.05 | 94.59 | **97.24** |
| | Ours | **89.23** | **93.82** | **95.16** | **97.33** |
| | Original PU | 76.79 | 80.63 | 85.53 | 88.43 |
| CIFAR-10 | RCGAN-U | 83.13 | 86.22 | 88.22 | **90.45** |
| | Ours | **87.64** | **87.92** | **88.60** | **90.69** |

To verify the advantage of theoretical property for our CNI-CGAN, we further compare it with RCGCN-U (Thekumparampil et al., 2018; Kaneko et al., 2019), the heuristic version of robust generation against unknown noisy labels setting without the theoretical guarantee of optimal condition. As observed in Table 1, our method outperforms RCGAN-U especially when the positive rate is low. When the amount of positive labeled data is relatively large, e.g., 10.0%, both our approach and RCGAN-U can obtain comparable performance.

**Visualization** To further demonstrate the superiority of CNI-CGAN compared with the other baselines, we present some generated images within $K+1$ classes from CGAN-A, RCGAN-U and CNI-CGAN on MNIST, and high-quality images from CNI-CGAN on Fashion-MNIST and CIFAR-10, in Figure 3. In particular, we choose the positive rate as 0.2% on MNIST, yielding the initial PU classifier with 69.14% accuracy. Given the noisy labels on all data, our CNI-CGAN can generate more accurate images of each class visually compared with CGAN-A and RCGAN-U. Results of Fashion-MNIST and comparison with CGAN-P on CIFAR-10 can refer to Appendix E.

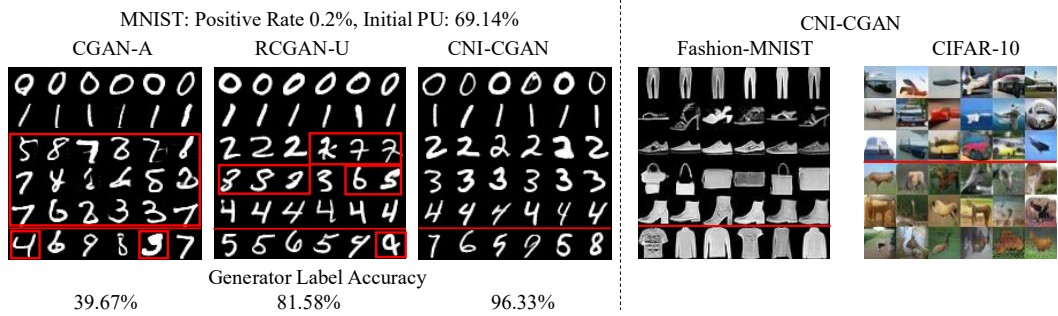

Figure 3: Visualization of generated samples on three datasets. Rows below the red line represent the negative class. We highlight the erroneously generated images with red boxes on MNIST.

## 4.2 ROBUSTNESS OF OUR APPROACH

**Robustness against the Initial PU accuracy** The auxiliary loss can help the CNI-CGAN to learn the clean data distribution regardless of the initial accuracy of PU classifiers. To verify that, we select distinct positive rates, yielding the pre-trained PU classifiers with different initial accuracies. Then we perform our method based on these PU classifiers. Figure 4 suggests that our approach can still attain the similar generation quality under different initial PU accuracies after sufficient training, although better initial PU accuracy can be beneficial to the generation performance in the early phase.

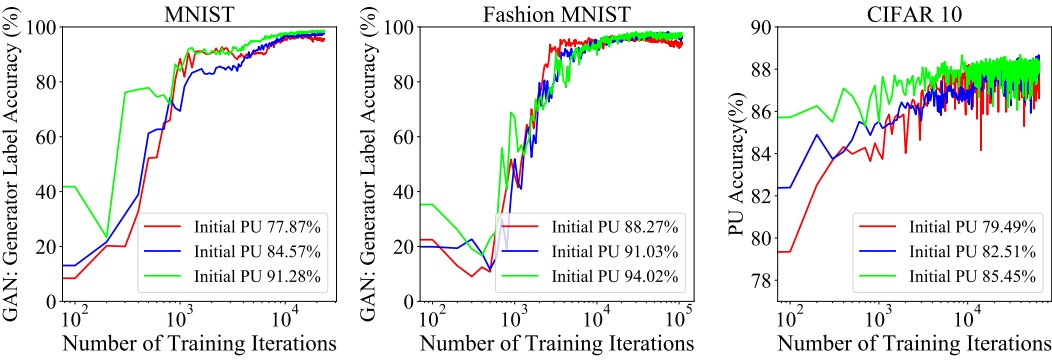

Figure 4: Tendency of generation performance as the training iterations increase on three datasets.

**Robustness against the Unlabeled data** In real scenarios, we are more likely to have little knowledge about the extra data we have. To further verify the robustness of CNI-CGAN against the unknown distribution of extra data, we test different approaches across different amounts and distributions of the unlabeled data. Particularly, we consider two different types of distributions for unlabeled data. Type 1 is $\left[\frac{1}{K+1}, ..., \frac{1}{K+1}, \frac{1}{K+1}\right]$ where the number of data in each class, including the negative data, is even, while type 2 is $\left[\frac{1}{2K}, ... \frac{1}{2K}, \frac{1}{2}\right]$ where the negative data makes up half of all unlabeled data. In experiments, we focus on the PU Accuracy to evaluate both the generation quality and the improvement of PU learning. For MNIST, we choose $1\%$ and $0.5\%$ for two settings while we opt for $0.5\%$ and $0.2\%$ on both Fashion-MNIST and CIFAR-10.

Figure 5 manifests that the accuracy of PU classifier exhibits a slight ascending tendency with the increasing of the number of unlabeled data. More importantly, our CNI-CGAN almost consistently outperforms other baselines across different amount of unlabeled data as well as distinct distributions of unlabeled data. This verifies that the robustness of our proposal to the distribution of extra data can be maintained potentially. We leave the investigation on the robustness against more imbalanced situations as future works.

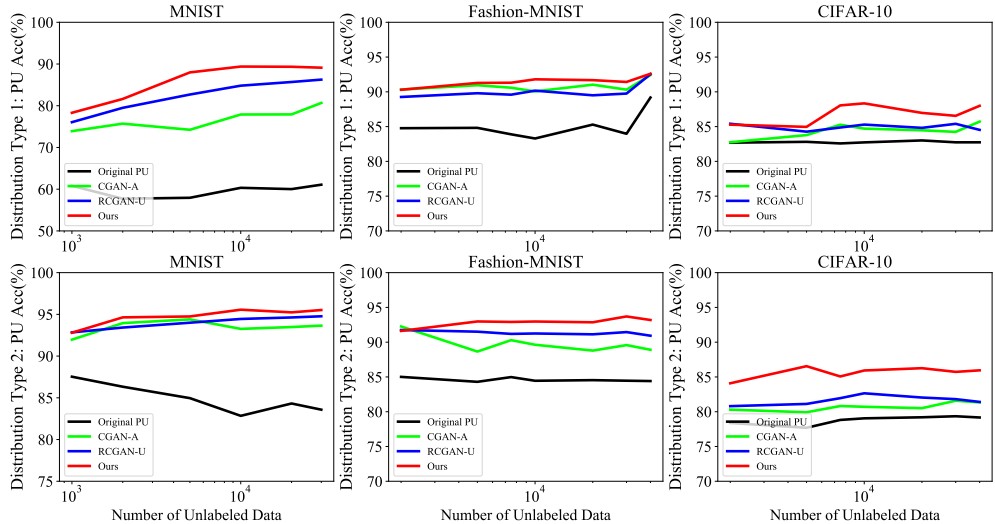

Figure 5: PU Classification accuracy of CGAN-A, RCGAN-U and Ours after joint optimization across different amounts and distribution types of unlabeled data.

## 5 RELATED WORKS

**Positive-Unlabeled (PU) Learning.**   Positive and Unlabeled (PU) Learning is the setting where a learner has only access to positive examples and unlabeled data (Bekker & Davis, 2020; Kiryo et al., 2017). One related work  (Hou et al., 2018) employed GANs (Goodfellow et al., 2014) to recover both positive and negative data distribution to step away from overfitting. Kato et al. (Kato et al., 2018) focused on remedying the selection bias in the PU learning. Besides, Multi-Positive and Unlabeled Learning (Xu et al., 2017) extended the binary PU setting to the multi-class version, therefore adapting to more practical applications. By contrast, our multi-positive unlabeled method absorbs the advantages of previous approaches, and in the meanwhile intuitively extends them to fit the differential deep neural networks optimization.

**Conditional GANs on Few Labels Data.**   To attain high-quality images with both fidelity and diversity, the training of generative models requires a large dataset. To reduce the need of huge amount of data, the vast majority of methods (Noguchi & Harada, 2019; Yamaguchi et al., 2019; Zhao et al., 2020) attempted to transfer prior knowledge of the pre-trained generator. Another branch (Lucic et al., 2019) is to leverage self- and supervised learning to add pseudo labels on the in-distribution unlabeled data in order to expand labeled dataset. Compared with this approach, our strategy can be viewed to automatically "pick" useful in-distribution data from total unknown unlabeled data via PU learning framework, and then constructs robust conditional GANs to generate clean data distribution out of predicted label noise. Please refer to more related works in Appendix B.

## 6 DISCUSSION AND CONCLUSION

In this paper, we proposed a new method, CNI-CGAN, to jointly exploit PU classification and conditional generation. It is, to our best knowledge, the first method of such kind to break the ceiling of class-label scarcity, by combining two promising yet separate methodologies to gain massive mutual improvements. CNI-CGAN can learn the clean data distribution from noisy labels given by a PU classifier, and then enhance the performance of PU classification through data augmentation in various settings. We have demonstrated, both theoretically and experimentally, the superiority of our proposal on diverse benchmark datasets in an exhaustive and comprehensive manner. In the future, it will be promising to investigate learning strategies on imbalanced data, e.g., cost-sensitive learning (Elkan, 2001), to extend our approach to broader settings, which will further cater to real-world scenarios where highly unbalanced data are commonly available. In addition, the leverage of soft labels in the design of CNI-CGAN is also promising.

**Ethics Statement.** Our designed CNI-CGAN framework can interplay with the PU classification and robust generation, which can mitigate the scarcity of class-labeled data. Leveraging extra data may correlate with the privacy issue as the privacy issue still exists in generative models. Thus, a privacy-guaranteed version of our algorithm can be further proposed in the future to handle the potential privacy issue.

**Reproducibility Statement.** For the theoretical part, we clearly state the related assumption and detailed proof process in Appendix A. In terms of the algorithm, our implementation is directly adapted from the public one of generative models and PU learning.

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

# A APPENDIX: PROOF OF THEOREM 1

Firstly, we recall some definitions. Denote $x_r$, $x_g$ as the real training and generated samples, respectively. $x$ are the population of all data, and $x_r$ are sampled from $p(x)$. $y_g$ represents the initial labels for the generator $G$, while $\tilde{y}$ indicates the labels perturbed by $\tilde{C}$ from $y_g$. The class-prior $\pi_i$ meets $\pi_i = P(y_g = i) = P(\mathcal{O}(x_r) = i)$. For a rigorous proof of Theorem 1, we elaborate it again in the appendix.

**Theorem 1** We assume that the following three mild assumptions can be met: (a) PU classifier is not overfitting on the training data, (b) $P(PU_\theta(x_g)|\mathcal{O}(x_g), y_g) = P(PU_\theta(x_g)|\mathcal{O}(x_g))$, (c) the conditional sample space is disjoint from each other class. Then,

(1) $P^g$ is a permutation matrix if the generator $G$ in CNI-CGAN is optimal, with the permutation, compared with an identity matrix, only happens on rows $\mathbf{r}$ where corresponding $\pi_r, r \in \mathbf{r}$ are equal.

(2) If $P^g$ is an identity matrix and the generator $G$ in CNI-CGAN is optimal, then $p^r(x, y) = p^g(x, y)$ where $p^r(x, y)$ and $p^g(x, y)$ are the real and generating joint distribution, respectively.

## A.1 PROOF OF (1)

*Proof.* For a general setting, the oracle class of $x_g$ given by label $y_g$ is not necessarily equal to $PU_\theta(x_g)$. Thus, we consider the oracle class of $x_g$, i.e., $\mathcal{O}(x_g)$ in the proof.

**Optimal $G$.** In CNI-CGAN, $G$ is optimal if and only if

$$p^r(x_r, PU_\theta(x_r)) = p^g(x_g, \tilde{y}). \tag{10}$$

The equivalence of joint probability distribution can further derive the equivalence of marginal distribution, i.e., $p^r(x_r) = p^g(x_g)$. We define a probability matrix $C$ where $C_{ij} = P(PU_\theta(x) = j|\mathcal{O}(x) = i)$ where $x$ are the population data. According to (c), we can apply $\mathcal{O}(\cdot)$ on both $x_r$ and $x_g$ in Eq. 10. Then we have:

$$
\begin{aligned}
P(\mathcal{O}(x_r) = i, PU_\theta(x_r) = j) &\overset{(c)}{=} P(\mathcal{O}(x_g) = i, \tilde{y} = j) \\
P(\mathcal{O}(x_r) = i)P(PU_\theta(x_r) = j|\mathcal{O}(x_r) = i) &= \\
\sum_{k=1}^{K+1} P(y_g = k, \mathcal{O}(x_g) = i)&P(\tilde{y} = j|y_g = k, \mathcal{O}(x_g) = i) \\
\pi_i C_{ij} &\overset{(a)}{=} \sum_{k=1}^{K+1} P(\mathcal{O}(x_g) = i|y_g = k)P(y_g = k)P(\tilde{y} = j|y_g = k) \\
\pi_i C_{ij} &= \sum_{k=1}^{K+1} P_{ik}^{g\top} \pi_k \tilde{C}_{kj},
\end{aligned}
\tag{11}
$$

where assumption (a) indicates that $PU_\theta(x_r)$ is close to $PU_\theta(x)$ so that $P(PU_\theta(x_r) = j|\mathcal{O}(x_r) = i) = P(PU_\theta(x) = j|\mathcal{O}(x) = i)$. Then the corresponding matrix form follows as

$$\Pi C = P^{g\top} \Pi \tilde{C} \tag{12}$$

**Definition.** According to the definition of $\tilde{C}$ and Law of Total Probability, we have:

$$
\begin{aligned}
P(y_g = i)P(PU_\theta(x_g) = j|y_g = i) &= \\
\pi_i \sum_{k=1}^{K+1} P(\mathcal{O}(x_g) = k|y_g = i)&P(PU_\theta(x_g) = j|\mathcal{O}(x_g) = k, y_g = i) \\
\pi_i \tilde{C}_{ij} &\overset{(b)}{=} \pi_i \sum_{k=1}^{K+1} P_{ik}^g P(PU_\theta(x_g) = j|\mathcal{O}(x_g) = k) \\
\pi_i \tilde{C}_{ij} &= \pi_i \sum_{k=1}^{K+1} P_{ik}^g C_{kj},
\end{aligned}
\tag{13}
$$

where the last equation is met as $p(x_g)$ is close to $p(x)$ when $G$ is optimal, and thus $P(PU_\theta(x_g) = j|\mathcal{O}(x_g) = k) = P(PU_\theta(x) = j|\mathcal{O}(x) = k)$. Then we consider the corresponding matrix form as follows

$$\Pi \tilde{C} = \Pi P^g C \tag{14}$$

where $\Pi$ is the diagonal matrix of prior vector $\pi$. Combining Eq. 14 and 12, we have $P^{g\top}\Pi P^g = \Pi$, which indicates $P^g$ is a general orthogonal matrix. In addition, the element of $P^g$ is non-negative and the sum of each row is 1. Therefore, we have $P^g$ is a permutation matrix with permutation compared with the identity matrix only happens on rows $\mathbf{r}$ where corresponding $\pi_r, r \in \mathbf{r}$ are equal. Particularly, if all $\pi_i$ are different from each other, then permutation operation will not happen, indicating the optimal conditional of $P^g$ is the identity matrix.

## A.2 PROOF OF (2)

We additionally denote $y_r$ as the real label of real sample $x_r$, i.e., $y_r = \mathcal{O}(x_r)$. According to the optimal condition of $G$ in Eq. 10, we have $p^r(x_r) = p^g(x_g)$. Since we have $P^g$ is an identity matrix, then $\mathcal{O}(x_g) = y_g$ a.e. Thus, we have $p^g(x_g|y_g = i) = p^g(x_g|\mathcal{O}(x_g) = i), \forall i = 1, .., K+1$. According the assumption (c) and Eq. 10, we have $p^r(x_r|\mathcal{O}(x_r) = i) = p^g(x_g|\mathcal{O}(x_g) = i)$. In addition, we know that $p^r(x_r|\mathcal{O}(x_r) = i) = p^r(x_r|y_r = i)$, thus we have $p^r(x_r|y_r = i) = p^g(x_g|y_g = i)$. Further, we consider the identical class-prior $\pi_i$. Finally, we have

$$
\begin{aligned}
p^r(x_r|y_r = i)\pi_i &= p^g(x_g|y_g = i)\pi_i \\
p^r(x_r|y_r = i)p(\mathcal{O}(x_r) = i) &= p^g(x_g|y_g = i)p(y_g = i) \\
p^r(x_r|y_r = i)p(y_r = i) &= p^g(x_g|y_g = i)p(y_g = i) \\
p^r(x_r, y_r) &= p^g(x_g, y_g).
\end{aligned}
\tag{15}
$$

$\square$

## B APPENDIX: MORE RELATED WORKS

**Positive-Unlabeled (PU) Learning.** Positive and Unlabeled (PU) Learning is the setting where a learner has only access to positive examples and unlabeled data (Bekker & Davis, 2020; Kiryo et al., 2017). One related work (Hou et al., 2018) employed GANs (Goodfellow et al., 2014) to recover both positive and negative data distribution to step away from overfitting. Kato et al. (Kato et al., 2018) focused on remedying the selection bias in the PU learning. Besides, Multi-Positive and Unlabeled Learning (Xu et al., 2017) extended the binary PU setting to the multi-class version, therefore adapting to more practical applications. By contrast, our multi-positive unlabeled method absorbs the advantages of previous approaches, and in the meanwhile intuitively extends them to fit the differential deep neural networks optimization.

**Conditional GANs on Few Labels Data.** To attain high-quality images with both fidelity and diversity, the training of generative models requires a large dataset. To reduce the need of huge amount of data, the vast majority of methods (Noguchi & Harada, 2019; Yamaguchi et al., 2019; Zhao et al., 2020) attempted to transfer prior knowledge of the pre-trained generator. Another branch (Lucic et al., 2019) is to leverage self- and supervised learning to add pseudo labels on the in-distribution unlabeled data in order to expand labeled dataset. Compared with this approach, our strategy can be viewed to automatically "pick" useful in-distribution data from total unknown unlabeled data via PU learning framework, and then constructs robust conditional GANs to generate clean data distribution out of predicted label noise.

**Robust GANs.** Robust Conditional GANs (Thekumparampil et al., 2018; Kaneko et al., 2019) were proposed to defend against class-dependent noisy labels. The main idea of these methods is to corrupt labels of generated samples before feeding to the adversarial discriminator, forcing the generator to produce sample with clean labels. Another supplementary investigation (Koshy Thekumparampil et al., 2019) explored the scenario when CGANs get exposed to missing or ambiguous labels, while another work (Chrysos et al., 2018) leveraged the structure of the model in the target space to address this issue. In contrast, the noises in our model stem from the prediction error of a given classifier. We employ the imperfect classifier to estimate the label confusion noise, yielding a new branch of Robust CGANs against "classifier" label noises.

**Semi-Supervised Learning (SSL).** One crucial issue in SSL (Miyato et al., 2018; Yu et al., 2019; Sun et al., 2019) is how to tackle with the mismatch of unlabeled and labeled data. *Augmented Distribution Alignment* (Wang et al., 2019) was proposed to leverage adversarial training to alleviate the bias, but they focused on the empirical distribution mismatch owing to the limited number of labeled data. Further, *Uncertainty Aware Self-Distillation* (Yanbei Chen, 2019) was proposed to concentrate on this under-studied problem, which can guarantee the effectiveness of learning. In contrast, our approach leverages the PU learning to construct the "open world" classification.

**Out-Of-Distribution (OOD) Detection** OOD Detection is one classical but always vibrant machine learning problem. PU learning can be used for the detection of outliers in an unlabeled dataset with knowledge only from a collection of inlier data (Hido et al., 2008; Smola et al., 2009). Another interesting and related

Table 2: Further evaluation of CGAN-P and Ours from the perspective of Inception Score on MNIST and Fashion-MNIST datasets.

| Positive Rates | | 0.75% | 1.0% | 3.0% | 5.0% | 10.0% |
|---|---|---|---|---|---|---|
| | | Inception Score ($\pm$ Standard Deviation) | | | | |
| MNIST | CGAN-P | 5.08±0.02 | 5.10±0.03 | 5.09±0.02 | 5.14±0.03 | 5.10±0.04 |
| | Ours | **5.60±0.01** | **5.59±0.02** | **5.65±0.02** | **5.52±0.01** | **5.63±0.02** |
| Fashion-MNIST | CGAN-P | 4.95±0.03 | 5.01 ± 0.03 | 5.04 ± 0.04 | 5.02±0.04 | 5.00 ±0.03 |
| | Ours | 4.99 ± 0.02 | 5.01 ± 0.02 | 5.03±0.01 | 5.07 ± 0.02 | 5.04 ± 0.02 |

work is *Outlier Exposure* (Hendrycks et al., 2018), an approach that leveraged an auxiliary dataset to enhance the anomaly detector based on existing limited data. This problem is similar to our generation task, the goal of which is to take better advantage of extra dataset, especially out-of-distribution data, to boost the generation.

**Learning from Noisy Labels** Rotational-Decoupling Consistency Regularization (RDCR) (Tsung Wei Tsai, 2019) was designed to integrate the consistency-based methods with the self-supervised rotation task to learn noise-tolerant representations. *Mutual Mean-Teaching* (Ge et al., 2020) was proposed to refine the soft labels on person re-identification task by averaging the parameters of two neural networks . In addition, the data with noisy labels can also be viewed as bad data. Another work (Guo et al., 2019) provided a worst-case learning formulation from bad data, and designed a data-generation scheme in an adversarial manner, augmenting data to improve the current classifier.

## C    APPENDIX: DETAILS ABOUT ALGORITHM 1

Similar in (Kiryo et al., 2017), we utilize the sigmoid loss $\ell_{\text{sig}}(t, y) = 1/(1 + \exp(ty))$ in the implementation of the PU learning. Besides, we denote $r_i = \widehat{R}_{\text{u}}^{-}\left(g; \mathcal{X}_{\text{u}}^i\right) - \pi_{\text{p}}\widehat{R}_{\text{p}}^{-}\left(g; \mathcal{X}_{\text{p}}^i\right)$ in the $i$-th mini-batch. Instructed by the algorithm in (Kiryo et al., 2017), if $r_i < 0$ we turn to optimize $-\nabla_\theta r_i$ in order to make this mini-batch less overfitting, which is slightly different from Eq. 4.

## D    APPENDIX: DETAILS ABOUT EXPERIMENTS

**PU classifier and GAN architecture** For the PU classifier, we employ 6 convolutional layers with different number of filters on MNIST, Fashion-MNIST and CIFAR 10, respectively. For the GAN architecture, we leverage the architecture of generator and discriminator in the tradition conditional GANs (Mirza & Osindero, 2014). To guarantee the convergence of RCGAN-U, we replace Batch Normalization with Instance Batch Normalization. The latent space dimensions of generator are 128, 128, 256 for the three datasets, respectively. As for the optimization of GAN, we deploy the avenue same as WGAN-GP (Gulrajani et al., 2017) to pursue desirable generation quality. Specifically, we set update step of discriminator as 1.

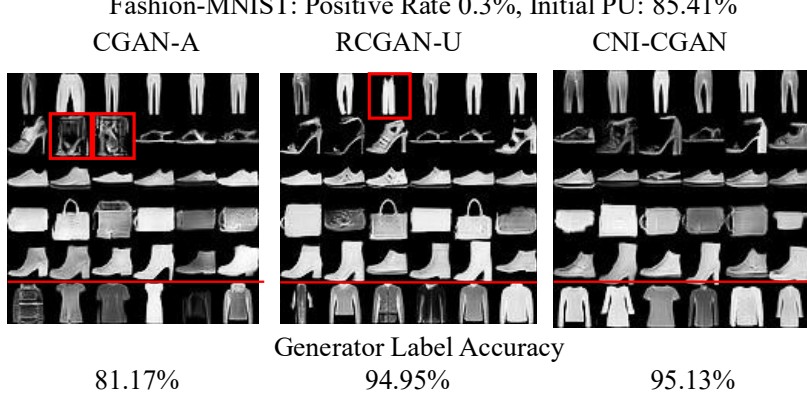

Figure 6: Visualization of generated samples from several baselines and ours on Fashion-MNIST.

CIFAR-10: Positive Rate 0.3%, Initial PU: 79.46%

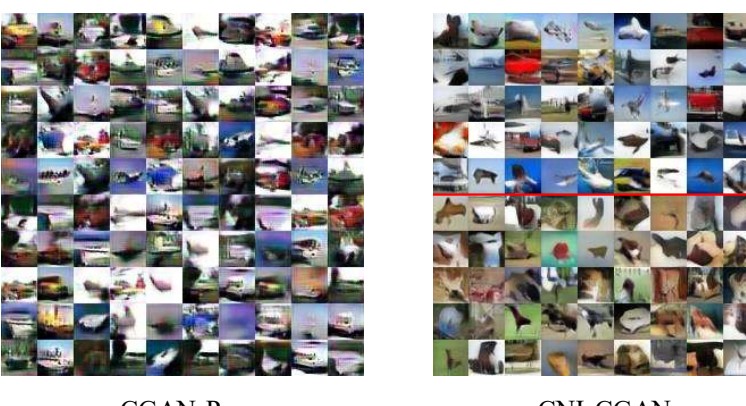

CGAN-P                                           CNI-CGAN

Figure 7: Visualization of generated samples from CGAN-P and ours on CIFAR-10.

**Choice of Hyper-parameters**   We choose $\kappa$ as 0.75, $\beta$ as 5.0 and $\lambda = 0.99$ across all the approaches. The learning rates of PU classifier and CGAN are 0.001 and 0.0001, respectively. In the alternate minimization process, we set the update step as 1 for PU classifier after updating the CGAN, and $L_0$ as 5 in Algorithm 1. We used the same and sufficient epoch for all settings (180 epochs for joint optimization) to guarantee the convergence as well as for fair comparisons.

**Further Evaluation of CGAN-P and Ours from the Aspect of Inception Score**   To better verify our approach can generate more pleasant images than CGAN-P, we additionally compare the Inception Score these two methods attain. Specifically, we trained a (almost) perfect classifier with 99.21 % and 91.33% accuracy for MNIST and Fashion-MNIST respectively. Then we generate 50,000 samples from the two approaches to compute Inception Score, the results of which are exhibited in Table 2. It turns out that our method attain the consistent superiority against CGAN-P on the Inception Score for MNIST, even though the generator label accuracy of these two approaches are comparable. Note that the two method obtains the similar Inception Score on Fashion-MNIST, but our strategy outperforms CGAN-P significantly from the perspective of generator label accuracy. Overall, we can claim that our method is better than CGAN-P.

# E   APPENDIX: MORE IMAGES

We additionally show some generated images on other datasets generated by baselines and CNI-CGAN, shown in Figure 6. Note that we highlight the erroneously generated images with red boxes. Specifically, on Fashion-MNIST our approach can generated images with more accurate labels compared with CGAN-A and RCGAN-U. Additionally, the quality of generated images from our approach are much better than those from CGAN-P that only leverages limited supervised data, as shown in Figure 7 on CIFAR-10.

