# OpenReview forum: "Classify and Generate Reciprocally: Simultaneous Positive-Unlabelled Learning and Conditional Generation with Extra Data"
_ICLR.cc/2022/Conference — ICLR 2022 Submitted_

### Official Review · Reviewer_jcSw · 2021-10-17

**Correctness:** 3
**Technical Novelty And Significance:** 3
**Empirical Novelty And Significance:** 2
**Recommendation:** 5
**Confidence:** 4

**Details Of Ethics Concerns:**

I do not have any ethics concerns on this paper.

**Main Review:**

Pros:
1. The high-level idea that simultaneously conduct PU classification and conditional generation is reasonable.
2. Experimental results show the effectiveness of the proposed method.

Cons:
1. The motivation is not that clear, especially why adopting multi-class PU learning rather than semi-supervised learning with out-of-distribution data.
2. Some necessary baseline methods should be compared.

**Summary Of The Paper:**

This paper proposes to leverage Positive-Unlabeled (PU) learning and the conditional generation with extra unlabeled data to simultaneously improve classification and generation performance. The authors show that their method is especially effective for handling out-of-distribution unlabeled data.

**Summary Of The Review:**

This paper proposes CNI-CGAN to jointly exploit PU classification and conditional generation, so as to address the label sparsity problem. I feel that the high-level motivation is reasonable, but I still have some concerns:
1. I think the motivation of this paper need more explanations. For the label scarcity problem mentioned in this paper, I feel that some recent semi-supervised learning methods are also applicable. For example, Semi-supervised learning under class distribution mismatch (AAAI 20), Safe deep semisupervised learning for unseen-class unlabeled data (ICML 20) also tackle the out-of-distribution data in semi-supervised learning. Besides, they are naturally deep model which do not need any extensions. In this sense, my question is, why adopting Multi-PU learning (needs extension to deep model) rather than these semi-supervised methods?
2. The authors wrote "In classification, a common setting to utilize unlabeled data is semi-supervised learning (Miyato et al., 2018; Sun et al., 2019; Berthelot et al.,2019), which usually assumes that the unlabeled and labeled data come from the same distribution, ignoring their distributional mismatch. In contrast, Positive and Unlabeled (PU) Learning (Bekker & Davis, 2020; Kiryo et al., 2017) is an elegant way of handling this under-studied problem, where a model has the only access to positive samples and unlabeled data." Firstly, as mentioned above, there are some recent semi-supervised methods which can well handle the distribution mismatch. Moreover, the authors claim that PU learning can elegantly handle this problem, which I cannot  fully agree. Note that in the setting of classical PU learning (e.g., uPU and nnPU), they also assume P data and U data come from the same underlying distribution, therefore I feel that the nnPU method deployed in this paper cannot well handle the  distribution mismatch problem.
3. For estimating the transition matrix \tilde{C}, usually it is a not an easy task in label noise learning. Some assumptions are usually needed, such as anchor point assumption (see Are Anchor Points Really Indispensable in Label-Noise Learning? NeurIPS 19), etc. However, I do not see such pre-set assumptions in this paper, so I doubt about the estimation quality for this matrix.
4. For experiments, many recent PU learning works are not compared. Besides, another important work "On Positive-Unlabeled Classification in GAN" (CVPR 20) should also be discussed and compared. Therefore, I think the empirical study should be enhanced.

---

> ### Author Response · Authors · 2021-11-15
> **Response**
>
> Thanks for your valuable comments. Below we address all your concerns. Please let us know if there are any further questions.
>
> **1.Discussion about SSL with OOD data.**
>
> We agree that some recent SSL under OOD data can still be applicable apart from the PU setting, but we believe the formulation of PU learning is more natural and straightforward to handle the issue when both positive and unlabeled data are accessible. For instance, SSL under the class distribution mismatch (AAAI 20) needs to introduce human-crafted regularization including Self-Distillation and OOD filtering, and Safe deep SSL for unseen-class unlabeled data (ICML 20) involves more complicated bi-level optimization. By contrast, our multi-class PU learning only involves the commonly-used cross-entropy loss, which is easier to understand and more straightforward. As you said, exploring the joint benefit of SSL with OOD and robust CGAN would serve as another promising direction, and we leave it as future works.
>
> **2. Assumption of PU learning.**
>
> Unlabeled data is the PU learning setting naturally contains negative data, and thus the PU classifier copes with the distribution match or OOD issue more naturally than SSL. The assumption of the same underlying distribution you mentioned is the joint distribution of Positive and Negative data, e.g., K+1 class joint distribution, which is a straightforward assumption that both uPU, nnPU and SSL with OOD are able to satisfy normally. We are glad to answer any further questions regarding this issue.
>
> **3. Assumption of Confusion Matrix.**
>
> A more related work is RCGAN [1] that pre-sets a known confusion matrix regarding noisy labels, which may not be applicable when exposing agnostic noisy labels in real applications. Our CNI-CGAN can be specifically robust against label noises coming from a classifier, and the key contribution is that we derived a non-trivial theorem to guarantee the convergence of CNI-CGAN to the clean data distribution. Moreover, we focus on the empirical superiority of CNI-CGAN to verify the estimation quality for the confusion matrix and note that a poor result by no means corresponds to a good estimation of the confusion matrix.
>
>
> **4. Discussion about more PU learning.**
>
> "On Positive-Unlabeled Classification in GAN" (CVPR 20) is a GAN-based work that treats generated images as unlabeled data for the discriminator, and thus incorporates the idea of PU learning for the discrimination classifier to further enhance the GAN's performance. The motivation and issue this paper involves and investigates are vastly different from our paper, although we understand some misunderstandings can be easily caused based on a similar paper title.
>
>
> [1] Kiran K Thekumparampil, Ashish Khetan, Zinan Lin, and Sewoong Oh. Robustness of conditional gans to noisy labels (NeurIPS 2018)

---

> > ### Comment · Reviewer_jcSw · 2021-11-25
> > **Reply to author rebuttal**
> >
> > I thank the authors for providing the responses to my previous concerns. I have carefully read their reply, and I'm not fully convinced regarding the 1st and 3rd questions.
> > For the necessity of adopting multi-PU learning (1st question), I do not think it is easier than semi-supervised learning. In terms of loss function, yes, it might be simpler. However, for the overall complexity in model designing, I do not think multi-PU learning is easier than semi-supervised learning.
> > For the assumption in estimating confusion matrix (3rd question), I'm still not very clear about why such assumption is not required. the authors claim the empirical studies show good performance, which I think is also not convincing.
> > Given the above reasons, I will keep my previous score.

---

> > > ### Author Response · Authors · 2021-11-25
> > > **Further Reply**
> > >
> > > We thanks for your reply.
> > >
> > > We agree that SSL with outliers would be another promising direction to handle this issue, but we think it is still complementary to multi-PU learning as at least we have demonstrated the efficacy of multi-PU classification in this paper to address this problem. As such, SSL with outliers can be more viewed in the extension part or as a potential alternative we can investigate in the future.
> > >
> > > Regarding the assumption, after reading the paper you mentioned, we think the confusion matrix analysis in RCGAN also satisfies the anchor point assumption, i.e., instances belonging to a specific class with probability exactly one or close to one, although RCGAN paper[1] did not explicitly mention this assumption. In particular, in our proof of Appendix A.1, we required this assumption naturally. Indeed, as you said, the anchor point assumption we satisfied can contribute to the estimation quality of the confusion matrix.

---

### Official Review · Reviewer_NqyD · 2021-11-01

**Correctness:** 3
**Technical Novelty And Significance:** 2
**Empirical Novelty And Significance:** 2
**Recommendation:** 5
**Confidence:** 3

**Main Review:**

## Pros

- The combination of positive-unlabeled classification and conditional generation with extra unlabeled data is naturally and sounds reasonable.
- The extensive experimental results indicate the effectiveness of the proposed framework on both classification and generation tasks.
- The writing is good, with a well-organized structure, clear illustration, rigorous equations.

## Cons

- Lack generation experiments on high-pixel datasets such as ImageNet.
- It should compare more SOTA PU and GAN methods.
- The proposed looks complex compared to the baseline methods, contains multiple components such as PU, G, C, D.



**Summary Of The Paper:**

This paper proposes a unified framework to leverage Positive Unlabeled classification and conditional generation with extra unlabled data and benefits the two tasks simultaneously.
It also proved the optimal condition of the proposed Classifier-Noise-Invariant Conditional GAN theoretically.


**Summary Of The Review:**

The proposed training framework which jointly targets PU classification and conditional generation is novel and sound.
But the novelty is limited and the lack some key experiments to show the improvements.
So I tend to reject the paper.

---

> ### Author Response · Authors · 2021-11-15
> **Response**
>
> We thank the reviewer's valuable remarks on our work. Below we address all your concerns. Please let us know if there are any further questions.
>
> **1.High-pixel datasets.**
>
> To the best of our knowledge, most works related to (robust) conditional generative models, e.g., RCGCN[1], normally test their models on MNIST, Fashion-MNIST, CIFAR-10 as their preference. This is mainly due to the difficulty of deploying generative models on ImageNet especially from the perspective of computational cost, which is also the common issue that the whole machine learning community is expected to work together rather than the unique one this paper needs to cope with specially. We believe the existing experiments on MNIST, Fashion-MNIST and CIFAR-10 in our paper are rigorous and sufficient followed by previously published papers[1], although we totally agree with you that verifying our method on ImageNet would definitely make our method more compelling. Thanks for this valuable suggestion again, and we will consider your suggestion in the final version of our paper.
>
> **2.More comparisons with PU and GAN.**
>
> The PU classifier we applied is based on non-negative risk[2] that is the most representative PU estimator that can mitigate overfitting and allows deep neural networks, and recent work[3] is normally based on [2]. Our framework can incorporate any PU classifier without any assumption on PU learning, and thus any SOTA PU estimator if that exists or appear in the future can be naturally within our CNI-CGAN framework. For the GAN architecture, we leverage the architecture of generator and discriminator in the traditional conditional GANs [4] and therefore our experimental results based on [4] are convincing and it is natural to expect the similar result by leverage of more advanced GAN models. We can add more experiments in the final version.
>
>
> **3.Model complexity and Novelty.**
>
> Our CNI-CGAN can also be viewed as a variant robust generative model, e.g., RCGAN[1] that has contained confusion matrix C into G, D GAN framework. Our main novelty and contribution are that our robust CGAN is robust against classifier-noise, implying that our proposed CNI-CGAN can learn the clean data distribution from the noisy labels predicted by an imperfect PU classifier. Therefore, we additionally incorporate PU into the design of robust CGAN in order to “imitate” the noise generation mechanism. Please refer to more details in Principle of Design of CNI-CGAN in Section 2.2. In summary, the classifier-noise-robust motivation determines the more complexity of our model architecture, but more importantly, we provide a **non-trivial** theorem to ensure the theoretical guarantee of CNI-CGAN even in this complex GAN architecture. This key contribution is expected to be recognized further.
>
>
> [1] Kiran K Thekumparampil, Ashish Khetan, Zinan Lin, and Sewoong Oh. Robustness of conditional gans to noisy labels (NeurIPS 2018)
>  [2] Ryuichi Kiryo, Gang Niu, Marthinus C du Plessis, and Masashi Sugiyama. Positive-unlabeled learning with a non-negative risk estimator. (NeurIPS 2017)
> [3] Su, Guangxin, Weitong Chen, and Miao Xu. "Positive-unlabeled learning from imbalanced data. (IJCAI 2021)
> [4] Mehdi Mirza and Simon Osindero. Conditional generative adversarial nets.

---

> > ### Comment · Reviewer_NqyD · 2021-11-25
> > **Response**
> >
> > Thanks to the author for the detailed response. I have read your comments carefully but am not convinced.
> > - For the dataset problem, as you said, it's difficult to deploy CGAN to the high-pixel dataset, which limits the usage of the proposed methods on the PU learning problem part.
> > - For the experiment comparison problem, I don't agree with your reply. "It is natural to expect the similar result by leverage of more advanced GAN models" is not the reason you only compare with Original PU and RCGAN-U. If you think it's natural, then you should show me the results.
> >
> > Based on the above reasons, I will keep my rating.

---

> > > ### Author Response · Authors · 2021-11-25
> > > **Further Reponse**
> > >
> > > Thanks for the further response.
> > >
> > > For the dataset problem, we truly understand your concern about the practicability of our method. However, we would argue that the limitation of GAN-based model on the high-pixel dataset is a **fundamental bottleneck** within the community. As our experimental implementation directly follows [1](NeurIPS 2018) that also mainly conducted experiments on MNIST and CIFAR-10, we contend that our demonstration is basically tenable, although we truly agree with you that the demonstration on the high-pixel dataset would definitely be more beneficial.
> > >
> > > We would like to add that the non-negative PU estimator [2] and CGAN[4] are the widely adopted PU classifier and conditional GANs, respectively, which we leveraged in our experiments and further made our conclusion. These are the first option we have, although demonstration on more PU estimators and CGAN models would be also useful, which we can add them in the appendix in the final version.

---

### Official Review · Reviewer_XLhs · 2021-11-03

**Correctness:** 3
**Technical Novelty And Significance:** 1
**Empirical Novelty And Significance:** 2
**Recommendation:** 1
**Confidence:** 4

**Main Review:**

The problem is practical and interesting. The paper generally is easy to follow and read.

However, my major concern is the novelty of the paper, which is a combination of PU and GAN. It is incremental with limited novelty.
There are GAN model for semi-supervised learning. How about comparing with them? This paper only compares with two methods.
What is the benefit of confusion matrix C to get noisy label?







**Summary Of The Paper:**

This paper aims to make full use of agnostic unlabeled data to improve classification and generation performance by leveraging Positive-Unlabeled (PU) classification and the conditional generation with extra unlabeled data. They evaluate on some benchmarks in different positive label ratio.

**Summary Of The Review:**

Based on the novelty, I tend to reject the paper.

---

> ### Author Response · Authors · 2021-11-15
> **Response**
>
> We sincerely thank the reviewer's efforts on the valuable comments. Below we address all your concerns. Please let us know if there are any further questions.
>
> **1. The novel proposal of CNI-CGAN rather than a simple combination of PU can CGAN.**
>
> We would like to clarify that the design of our Classifier-Noise-Invariant~(CNI) CGAN is novel rather than a simple combination of PU learning and CGAN. The key difference is that we incorporate PU learning into the robust CGAN framework, and after a non-trivial design in Figure 1 we eventually figure out a novel and useful **Classifier-Noise-Invariant** robust CGAN with a **non-trivial** theoretical guarantee. The simple combination can not achieve this contribution by any means.
>
> Specifically, existing label-noise-robust CGANs typically assume a known label noise confusion matrix or heuristically learn it without any theoretical guarantee, such as RCGAN-U. By contrast, the label noise in our CNI-CGAN is directly determined by an imperfect (PU) classifier rather than a given label noise matrix, as shown in Figure 1. More importantly, we provide a **non-trivial** theoretical convergence guarantee of CNI-CGAN to the clean data distribution, serving as our main contribution in this paper. Therefore, the design of our CNI-CGAN is different from existing settings instead of a simple combination of PU learning and CGAN.
>
> **2.GAN models for Semi-supervised Learning.**
>
> Semi-supervised learning (SSL) normally assumes that the unlabeled data has the identical data distribution as the labeled data, e.g., [1], and thus we can design a delicate GAN architecture to handle this issue. However, PU learning that our paper considers is vastly different from SSL setting, where the unlabeled data indeed contains out-of-distribution data, yielding a K+1 classification task. Thus, due to existing GAN methods in SSL are not applicable to our setting, we specifically compare with two natural baselines in our PU setting, including CGAN-P and CGAN-A that respectively train CGAN on K-class positive data and all K+1-class data labeled by the PU classifier. Experimental results demonstrate the effectiveness of our CNI-CGAN in our setting.
>
> **3.The role of Confusion Matrix.**
>
> The confusion matrix is a typical option in the design of robust GAN models, e.g., [2, 3] in order to model the noise generation mechanism from the noisy labels. Existing works [2, 3] assume a known label noise confusion matrix or heuristically learn it without any theoretical guarantee, while our CNI-CGAN directly models the noisy labels determined by the classifier with a strong theoretical guarantee. Please refer to more details in Principle of the design of CNI-CGAN in Section 2.2.
>
> In summary, we propose a novel robust classifier-aware CGAN called CNI-CGAN that is able to learn the clean data distribution on all unlabeled data, including OOD data, with noisy labels provided by the PU classifier. The theoretical guarantee of CNI-CGAN makes our method more practical to cope with real scenarios when we are only accessible to little class-labeled data, and extra unlabeled data. To the best of our knowledge, our work is novel to handle this issue from the methodology perspective.
>
> [1] Dai, Zihang, et al. Good semi-supervised learning that requires a bad gan. (NIPS 2017)
> [2] Kiran K Thekumparampil, Ashish Khetan, Zinan Lin, and Sewoong Oh. Robustness of conditional gans to noisy labels (NeurIPS 2018)
> [3] Takuhiro Kaneko, Yoshitaka Ushiku, and Tatsuya Harada. Label-noise robust generative adversarial networks. (CVPR 2019)

---

> ### Author Response · Authors · 2021-11-28
> **Please give your further opinions on our paper**
>
> Dear Reviewer XLhs,
>
> We have updated our manuscript and replied to your comments. Would you please check whether our efforts are satisfactory and raise your score? There is only one day left. Many thanks!
>
> Authors

---

### Official Review · Reviewer_EFMY · 2021-11-03

**Correctness:** 4
**Technical Novelty And Significance:** 3
**Empirical Novelty And Significance:** 3
**Recommendation:** 5
**Confidence:** 5

**Main Review:**

The pros and cons of the paper are described below.

## Pros ##
Bridging different generalization methods is a common technique to improve the performance of deep learning, as verified by much previous work. However, it seems to be the fact that utilizing PU and conditional generation is the unique contribution of this work. Such a new perspective does provide new insights to the community and can motivate future research in the field of semi-supervised learning. Meanwhile, combing PU and conditional generation is not an easy task. To this end, the authors designed the model of CNI-CGAN, which introduces several novel designs. The theoretical proof also seems reasonable.

## Cons ##
For the empirical evaluation part, the authors only verified their method on several relatively small datasets like MNIST and CIFAR. However, the main claim of this paper is that they can relive the massive labeled data consumption of training deep networks. Thus, it is important to verify the method on some relatively large datasets like ImageNet. The reviewer understands that there may not be a proper data source of unlabeled data or an out-of-distribution data source for ImageNet. However, this fact also suggests that the semi-supervised learning method this paper puts forward may not be easily implemented in practice.


**Summary Of The Paper:**

This paper targets at relieving the massive labeled data consumption of deep learning through the framework of semi-supervised learning. In particular, it finds out that two training approaches, Positive-Unlabeled classification and the conditional generation, can benefit each other. Jointly conducting these two approaches can push better performance on both tasks, thus eventually achieving better performance with a limited amount of labeled data. The authors combined the two tasks with a new type of GAN network. They further gave the corresponding theoretical proof for this new GAN model and verified its performance on the benchmark datasets.

**Summary Of The Review:**

Considering the advantages and disadvantages of this paper, the reviewer recommends weak accept. If the authors can verify their method on ImageNet, the reviewer will raise the rating.

After reading the authors' rebuttal as well as the other reviews, I lower my rating because I don't see the experimental results on ImageNet as my expectation.

---

> ### Author Response · Authors · 2021-11-15
> **Response**
>
> Thanks for appreciating our work.
>
> To the best of our knowledge, most works related to (robust) conditional generative models, e.g., RCGCN[1], normally test their models on MNIST, Fashion-MNIST, CIFAR-10 as their preference. This is mainly due to the difficulty of deploying generative models on ImageNet especially from the perspective of computational cost, which is also the common issue that the whole machine learning community is expected to work together rather than the unique one this paper needs to cope with specially. We believe the existing experiments on MNIST, Fashion-MNIST and CIFAR-10 in our paper are rigorous and sufficient followed by previously published papers[1], although we totally agree with you that verifying our method on ImageNet would definitely make our method more compelling. Thanks for this valuable suggestion again, and we will consider your suggestion in the final version of our paper.
>
> [1] Kiran K Thekumparampil, Ashish Khetan, Zinan Lin, and Sewoong Oh. Robustness of conditional gans to noisy labels (NeurIPS 2018)

---

> ### Author Response · Authors · 2021-11-28
> **Please give your further opinions on our paper**
>
> Dear Reviewer EFMY,
>
> We have replied to your comments. Would you please check whether our efforts are satisfactory and raise your score? There is only one day left. Many thanks!
>
> Authors

---

### Comment · Area_Chair_DB5m · 2021-11-22
**External discussion**

(note that this is **not** the internal discussion)

Hi all reviewers,

Can you take a look at the rebuttal and let the authors know that you have done so?

AC

---

### Decision · Program_Chairs · 2022-01-20

**Decision:**

Reject

**Comment:**

The paper combines discriminative and generative positive-unlabeled learning into a single framework. The reviewers argued the novelty and contributions are not enough for ICLR and unfortunately we cannot accept it for publication.